# Child Disruptions, Remote Learning, and Parent Mental Health during the COVID-19 Pandemic

**DOI:** 10.3390/ijerph19116442

**Published:** 2022-05-25

**Authors:** Stephanie Deeb, Devin Madden, Timnit Ghebretinsae, Joyce Lin, Umut Ozbek, Victoria Mayer, Nita Vangeepuram

**Affiliations:** 1Icahn School of Medicine at Mount Sinai, New York, NY 10029, USA; stephanie.deeb@icahn.mssm.edu (S.D.); devin.madden@mountsinai.org (D.M.); timnit.ghebretinsae@mountsinai.org (T.G.); victoria.mayer@mountsinai.org (V.M.); 2Department of Population Health Science and Policy, Icahn School of Medicine at Mount Sinai, New York, NY 10029, USA; joyceleaves@gmail.com (J.L.); umut.ozbek@mountsinai.org (U.O.); 3Institute for Health Equity Research, Icahn School of Medicine at Mount Sinai, New York, NY 10029, USA; 4Department of Environmental Medicine and Public Health, Icahn School of Medicine at Mount Sinai, New York, NY 10029, USA

**Keywords:** COVID-19, mental health, parents, education, distance, anxiety, depression, trauma and stressor-related disorders, stress disorders, post-traumatic

## Abstract

New York City (NYC) was an epicenter of the COVID-19 pandemic, which resulted in broad economic, social, and emotional consequences in the lives of individuals. The current study examined associations between pandemic-related stressors and adverse mental health symptoms among NYC parents/caregivers. Community-based participatory research was used to develop a survey, and logistic regression models were utilized to assess associations between factors including disruptions in child routines and remote learning, and parent-reported symptoms of stress, anxiety, depression, and post-traumatic stress disorder (PTSD). Some 91.0% of parents reported stress and 41.2, 26.6, and 33.7% reported symptoms of anxiety, depression, and PTSD, respectively. Most parents (87.6%) reported cancellation of at least one child activity. Of the parents, 60.3% reported that their children participated in remote learning and the majority (70.3%) reported feeling overwhelmed by it. Having more cancelled child activities was associated with higher odds of reported mental health symptoms, with not being able to play outside associated with higher odds of anxiety (1.80 (1.26, 2.58), *p* = 0.001), depression (1.93 (1.29, 2.91), *p* = 0.002), PTSD (1.64 (1.13, 2.39), *p* = 0.009), and stress (2.34 (1.27, 4.44), *p* = 0.008). Feeling overwhelmed by remote learning was also associated with higher odds of all four outcomes. Pre-existing mental illness, lower resilience scores, and lower socioeconomic status emerged as additional factors associated with symptoms of mental illness. These findings highlight the importance of resources to minimize adverse psychological effects among vulnerable families.

## 1. Introduction

The COVID-19 pandemic has had a global impact, including major consequences in cities across the United States. During the first wave of the pandemic in New York City (NYC), from March–April 2020, confirmed COVID-19 cases peaked at over 6500 daily, while hospitalization rates soared to 2000 per day with over 800 deaths daily [1]. The NYC Department of Health and Mental Hygiene (DOH) reported more than 200,000 cases of COVID-19 between March and June 2020, along with over 18,000 confirmed and probable COVID-19-associated deaths [2,3]. In February 2021, infection rates spiked again at nearly 8000 daily new cases, reflecting the high burden of disease experienced by the NYC population during both the first and second waves of the pandemic [1].

NYC’s economy also suffered. Fifty one percent of households in the NY Metro area, surveyed between April and October 2020, reported income loss since March 2020; this rate was even higher (60%) among households with children [4]. As a result, parents reported having difficulty paying for housing and food; parents with lower incomes, in particular, were less likely to be able to work from home and reported difficulty coordinating childcare during the pandemic [5]. Public schools across NYC switched to remote learning in March 2020, with the NYC Department of Education reporting that 85% of students engaged in remote learning in the spring of 2020, and most schools remained largely remote for the remainder of the year [4,6]. Prior studies have described challenges inherent in transitioning to an online learning platform, including difficulties optimizing ergonomic factors to create an engaging learning environment for children [7]. Additionally, prior research has highlighted inequities in access to and the utilization of digital technologies that are critical in supporting online learning; these disparities have been associated with socioeconomic, geographical, and cultural factors [8].

The COVID-19 pandemic and the associated risks of disease, economic changes, and life disruptions have the potential to leave lasting psychological impacts upon individuals in affected communities. Research examining past natural disasters and pandemics has highlighted these psychological effects; while most individuals exposed to a disaster recover effectively, many experience mental health challenges and symptoms of mental illness including anxiety, depression, and post-traumatic stress disorder (PTSD) [9,10,11]. Particularly during the COVID-19 pandemic, stressors including social isolation during quarantine, fears of infection and illness, and financial challenges may contribute to the development of mental illness [12,13]. Indeed, several studies have indicated higher levels of psychological distress, anxiety, and depression among adults in the general population following the pandemic [14,15,16,17,18]. Nearly half of adult New Yorkers reported experiencing symptoms of anxiety related to COVID-19 and over one-third reported symptoms of depression, according to a survey administered by the NYC DOH in April–May 2020 [19]. In particular, studies have reported worsening mental health among parents compared to those living without children in the home, along with high stress levels and high rates of anxiety and depression among surveyed parents [20,21,22]. According to the NYC DOH, over one-third of surveyed parents reported a negative impact on their child’s emotional or behavioral health, contributing to the concerns parents faced during the pandemic [19].

No prior study has examined the impacts of these pandemic-related stressors upon the symptoms of mental illness among parents in NYC, spanning the first two major waves of the pandemic. In this study, we used community-based participatory research (CBPR) to virtually plan, develop, and implement a survey examining the pandemic’s impact on NYC communities, including the effects of disruption in child routines on parental mental health, with the goal of understanding specific factors that may be associated with certain mental health outcomes. In particular, we hypothesized that pandemic-related disruptions in child routines, pre-existing mental illness, and lower socioeconomic status would be associated with adverse psychological outcomes among parents in NYC during the pandemic.

## 2. Materials and Methods

### 2.1. The Speak Up on COVID Survey

The study was conducted in accordance with the Declaration of Helsinki and approved as an exempt human research study by the Institutional Review Board of the Icahn School of Medicine at Mount Sinai. The Speak Up on COVID survey was developed through collaboration with more than 100 community and academic partners. The Mount Sinai research team worked with partners to identify survey domains, including demographics; medical history; experiences during the pandemic around testing, diagnosis, hospitalization, and access to care; socioeconomic impacts; mental health; social determinants of health; and prevention and lifestyle behaviors. We piloted the survey with partners and translated it into ten languages commonly spoken in NYC. The survey was finalized and launched in May of 2020.

### 2.2. Participants

Our community partners were vital in implementing a convenience sampling outreach and recruitment strategy. Each collaborating organization shared the survey through newsletters, social media, and, if possible, in person. We continuously modified our outreach strategy in order to improve the size and diversity of our sample; a team of student volunteers conducted outreach in person as local safety guidelines allowed, through traditional and social media, and in targeted neighborhoods with high COVID prevalence. We offered an incentive, through a raffle, to participants for completing the survey. The goal of the Speak Up on COVID survey was to understand the broad range of impacts of the pandemic on NYC residents who were disproportionately affected by COVID-19; we chose a convenience sampling strategy in order to collaborate with our community partners to reach neighborhoods most impacted by COVID-19. As the study was IRB exempt, full consent was not required and instead we presented an IRB-approved research information sheet with the study details, which all participants had to review prior to participation and attest that they met eligibility criteria. Eighteen hundred and fifty-four adults in NYC completed the survey between May 2020 and April 2021, and this manuscript includes data from 523 respondents who had children under 18 years old in their home. 

### 2.3. Outcomes

Outcomes included parents’ reported symptoms of anxiety, depression, post-traumatic stress disorder (PTSD), and stress. Anxiety and depression symptoms were captured using the PHQ-4; scores for each condition ranged from 0 to 6, and participants were considered to screen positive if their scores were greater than or equal to 3 [23]. PTSD was evaluated using the PC-PTSD-5; scores ranged from 0 to 5, and participants were considered to have PTSD if their scores were greater than or equal to 3 [24]. Stress was measured using the following question: “Stress is when someone feels tense, nervous, anxious, or can’t sleep at night because their mind is troubled. How stressed are you?”, and participants were considered to demonstrate stress if they responded with “a little bit”, “somewhat”, “quite a bit”, or “very much” [25].

### 2.4. Predictors

The primary predictor was disruption of child routines, measured by the quantity and types of disruptions children experienced, including cancellation of after school activities, not being able to play outside or with other children, greater use of TV or internet for entertainment, participation in remote learning, and whether parents felt prepared or overwhelmed to teach their children at home. Table 1 demonstrates these items as well as the survey question used to obtain this information [26].

Other covariates included sociodemographic factors such as parent gender, age, race/ethnicity, income, and education. Gender was identified through a check all that apply list that included man, woman, non-binary, gender non-conforming, transgender, agender, and other. In this analysis, we compared those who selected only man with those who selected only woman. Those who selected more than one gender identity were considered “other gender”. Age was categorized as 18–24, 25–44, 45–64, and 65+ years old. For race/ethnicity, if respondents self-identified as Hispanic, they were classified as Hispanic regardless of selected race/s. If respondents did not self-identify as Hispanic and only selected white race, they were classified as non-Hispanic white; these rules applied to non-Hispanic Black, non-Hispanic Asian, prefer not to say, and other (including Hawaiian/Pacific Islander, American Indian/Alaskan native, and other races). If participants self-identified as non-Hispanic but selected more than one race, they were classified as other/mixed race. Income was categorized as <20 k, 20–40 k, 40–60 k, 60–80 k, 80 k+ USD, and unknown. Education was categorized as high school/GED or less, some college, bachelor’s degree, and postgraduate; those who responded “prefer not to say” were excluded from the analyses of education (Table 1).

Additional predictors included resilience score, social support, and pre-existing mental illness, which was assessed using a question about medical history: “Have you ever been told by a doctor or nurse that you have: (Please check all that apply)”, with a checklist of different conditions including anxiety, depression, PTSD, and other mental health conditions [27]. Resiliency scores were calculated based on the CD-RISC2 scale (possible range 0 to 8, with higher scores indicating greater resilience) [28]. Social support was evaluated through the question, “How often do you see or talk to people that you care about and feel close to?” with possible choices including less than once per week, 1–2 times, 3–4 times, >5 times per week, or “prefer not to say” (Table 1).

### 2.5. Statistical Analysis

Summary statistics were calculated to describe participant characteristics and outcomes. Variables were checked for normality and linearity before the implementation of univariable and multivariable logistic regression models. Mean and standard deviation or median and interquartile range (IQR) were calculated for continuous variables, and frequency with proportion were calculated for categorical variables.

We fitted univariable and multivariable logistic regression models to assess the associations between disruption in child routines and parental mental health symptoms. Predictors which were significant in the univariable regression models were added to the multivariable regression models. In the multivariable models, education level, rather than income, was included as an indicator of socioeconomic status to prevent multi-collinearity; both covariates may be used as markers of socioeconomic status, however, education level was used in this case as it was somewhat more evenly distributed across our study sample. All statistical analyses were performed using R 3.6.0 (Vienna, Austria).

## 3. Results

Between May 2020 and April 2021, 1854 adults in NYC completed the Speak Up on COVID survey; 523 respondents had children under 18 years old in the home. The descriptive statistics of this sample are provided in Table 1. The sample was predominantly women, and half of the respondents were between 25–44 years of age. The sample was 36.5% non-Hispanic white, 14.7% non-Hispanic Black, 27.3% Hispanic, 14.7% non-Hispanic Asian, and 6.7% non-Hispanic other/mixed race. Half of respondents reported an income of USD 80,000 or higher, and the majority had a bachelor’s or postgraduate-level degree (27.6% and 41.9%, respectively; Table 1).

Table 1 reports primary outcomes, including the prevalence of mental health symptoms. Overall, stress was most prevalent, and nearly half of the sample reported symptoms of anxiety, nearly one third reported symptoms of depression, and one third reported symptoms of PTSD. Most parents reported cancellation of at least one child activity, with nearly half reporting cancellations of 3–4 activities or programs and more than half reporting cancellation of after school activities and greater use of the internet or TV as entertainment. Over half of the sample reported that their children were unable to go outside to play, and almost three quarters of the respondents reported that their children were unable to play with other children. Additionally, more than half of parents surveyed reported that their children participated in remote learning; among this subset of parents, less than half reported feeling prepared to supervise remote learning, and the majority reported feeling overwhelmed by this task (Table 1).

The median resiliency score among the overall sample was 6 (IQR 5, 8), with the majority (82.3%) receiving social support at least once per week. Overall, nearly one-third of the sample reported pre-existing mental illness prior to the pandemic (Table 1).

### 3.1. Disruption in Child Routines and Parent Mental Health Symptoms

The results of the univariable analyses are presented in Table 2 and Table 3, with multivariable results provided in Appendix A.

As the number of cancelled child activities increased, there were statistically significant increases in the odds of parent-reported anxiety and PTSD symptoms in a dose–response relationship. This association was not significant for depressive symptoms and stress (Table 2).

Several specific disruptions in child routines were associated with parent-reported mental health symptoms. Cancellation of after school activities was associated with higher odds of parental PTSD symptoms, while greater use of the TV or internet to entertain children was associated with higher odds of parent stress (Table 2). Disruptions in children’s play routines were associated with several parent-reported mental health symptoms; not being able to play with other children was associated with higher odds of anxiety and PTSD symptoms, and not being able to go outside to play was associated with higher odds of all four mental health outcomes (Table 2). In the adjusted analysis, not being able to play with other children remained associated with higher odds of anxiety (OR (95% CI) 1.83 (1.08, 3.13), *p* = 0.025), and not being able to play outside remained associated with higher odds of stress (2.21 (1.04, 4.83), *p* = 0.042; Appendix A).

Participation in remote learning was not associated with significantly higher odds of parent-reported mental health symptoms (Table 2). However, among the parents of children who participated in remote learning, feeling prepared to supervise remote learning was associated with lower odds of anxiety, depression, PTSD, and stress. Additionally, feeling overwhelmed by remote learning was associated with higher odds of all four adverse mental health outcomes among parents (Table 2). In the adjusted analysis, feeling overwhelmed by remote learning remained associated with higher odds of anxiety (2.38 (1.18, 4.94), *p* = 0.017; Appendix A).

### 3.2. Individual Factors and Parent Mental Health Symptoms

Compared to non-Hispanic white race and ethnicity, Hispanic ethnicity was associated with higher odds of parent-reported depression symptoms, while non-Hispanic Asian parents were less likely to report stress. The only significant association with age was that parents aged 45–64 had lower odds of reported depression symptoms than parents aged 18–24, and there were no significant associations between parent gender and mental health symptoms (Table 3). In multivariable analyses, no associations between race/ethnicity and parent mental health symptoms remained significant (Appendix A).

As income level increased, the odds of reported PTSD symptoms decreased. Additionally, an annual household income of USD 80,000 or higher was associated with lower odds of anxiety, depression, and PTSD compared to an income of less than USD 20,000 (Table 3). Similarly, as parent education level increased, there were significantly lower odds of anxiety, depression, and PTSD in a dose–response relationship (Table 3). No associations were observed between income or education and parent stress. In multivariable analyses, some amount of college-level education remained associated with lower odds of anxiety, and postgraduate education remained associated with lower odds of depression (0.47 (0.22, 0.99), *p* = 0.048, and 0.29 (0.13, 0.64), *p* = 0.002, respectively; Appendix A). Among the subset of parents whose children engaged in remote learning, these adjusted associations were similar to those observed in the overall sample (0.35 (0.13, 0.9), *p* = 0.033 and 0.24 (0.08, 0.73), *p* = 0.0120, respectively; Appendix A).

Greater resiliency was associated with lower odds of parent mental health symptoms, with one unit increase in resiliency score associated with lower odds of anxiety, depression, PTSD, and stress (Table 3). These associations remained significant in the adjusted analysis (0.83 (0.73, 0.93), *p* = 0.002; 0.73 (0.63, 0.84), *p* < 0.001; 0.76 (0.66, 0.86), *p* < 0.001; 0.56 (0.38, 0.77), *p* = 0.001, respectively; Appendix A). In the adjusted analysis among parents whose children engaged in remote learning, a higher resiliency score was associated with lower odds of depression, PTSD, and stress (0.72 (0.59, 0.87), *p* = 0.001; 0.81 (0.67, 0.97), *p* = 0.022; 0.82 (0.7, 0.95), *p* = 0.008, respectively; Appendix A).

More frequent social support was generally associated with lower odds of mental health symptoms among parents. Parents who reported receiving social support 3–5 times or more than 5 times per week had lower odds of depression and stress, and those who received social support more than 5 times per week also had lower odds of PTSD symptoms compared to parents receiving social support less than once a week (Table 3). In the adjusted analysis, the associations between social support and depression and PTSD were no longer significant. However, any amount of social support (1–2 times, 3–4 times, and more than 5 times per week), compared to parents receiving social support less than once a week, was associated with lower odds of stress in the multivariable analyses (0.12 (0.01, 0.66), *p* = 0.045; 0.12 (0.01, 0.65), *p* = 0.045; 0.10 (0.01, 0.53), *p* = 0.029; 0.03 (0.00, 1.13), *p* = 0.043, respectively; Appendix A). In the adjusted analysis among parents whose children engaged in remote learning, no associations between social support and parental mental health symptoms remained significant (Appendix A).

A pre-existing mental health diagnosis among parents was associated with significantly higher odds of reported anxiety, depression, and PTSD symptoms during the pandemic (Table 3); these associations remained significant in adjusted analyses in the overall sample (3.13 (2.04, 4.84), *p* < 0.001; 3.28 (2.02, 5.36), *p* < 0.001; 4.79 (3.06, 7.59), *p* < 0.001, respectively; Appendix A) and the subsample of parents whose children engaged in remote learning (3.79 (2.05, 7.16), *p* < 0.001; 4.81 (2.41, 9.86), *p* < 0.001; 4.58 (2.45, 8.72), *p* < 0.001, respectively, as well as stress 4.57 (2.6, 8.14), *p* < 0.001; Appendix A).

## 4. Discussion

### 4.1. Mental Health Outcomes among Parents

Adverse mental health symptoms were prevalent among parents in our study sample, with stress being particularly widespread. Our findings are consistent with research investigating mental health outcomes among parents following disasters and public health crises, including studies of COVID-19. Prior work has demonstrated adverse psychological outcomes in the wake of a disaster, including anxiety, depression, and PTSD, among other mental and behavioral disorders [9,10,11]. These studies cited having children among several factors predicting a higher risk of negative mental health outcomes [9,10,11,29]. Investigations of the COVID-19 pandemic, in particular, have highlighted children in the home as a predictor of psychological distress, anxiety, and depression among adults, along with worsening mental health and increased frequency of negative mood reported among surveyed parents [4,15,19,20,21,30,31]. Previous studies indicate the comparable prevalence of these outcomes among NYC populations, with symptoms of anxiety and/or depression reported among one-third of surveyed adults living with children in the NY Metro area [4]. Stressors including parenting responsibilities, changes in childcare or employment, financial burdens, and concerns for their children’s health may contribute to the considerable risk of negative mental health outcomes among parents [10,11,13,29,32].

Among our sample of parents, symptoms of negative psychological outcomes were widespread; however, several pandemic-related and individual factors appeared to confer greater risk or a protective effect upon the development of these outcomes; these findings highlight potential vulnerable groups and targetable areas for interventions geared towards supporting parental mental health through pandemic-related changes.

### 4.2. Disruptions in Child Routines and Remote Learning

Disruptions in routines were widely reported among parents; most endorsed at least one disruption in their child’s life due to the pandemic, with specific changes including cancellation of in-person activities, less playtime outdoors, and less playtime with other children, in exchange for greater consumption of TV and internet. There was an association between disruption in child routines and adverse parental mental health outcomes, with a higher number of cancelled activities associated with higher odds of anxiety, PTSD, and stress reported by parents; this association was not seen for symptoms of depression, possibly due to the lower overall prevalence of depression among the sample. In particular, not being able to play with other children or play outside were both associated with greater odds of anxiety among parents. In addition, the majority of parents indicated a switch to remote learning for their children, with most reporting feeling unprepared and overwhelmed by this change; feeling unprepared or overwhelmed by remote learning were both independently associated with higher odds of parent mental health sequelae.

Similar to our study, prior research has suggested that pandemic-related disruptions in child routines, particularly the switch to remote learning, posed demands upon parents’ time and finances to adapt to new practices and support their children’s education [33,34]. Parent-reported stress or depression has been negatively associated with perceived preparation to supervise remote learning [20,35], with many parents reporting barriers including lack of access to the computing hardware and internet required to support remote education, particularly among parents with lower incomes [36]. These findings highlight the importance of adequate funding and educational resource allocation when remote learning and social isolation practices are necessary, and can help guide policy makers to ensure that pandemic-related policies in the education sector address the provision of resources to parents in greatest need of support for online learning. Additionally, parents have reported impacts of homeschooling including isolation and decreased quality of education [20,35]; furthermore, socialization has been demonstrated to be critical for child development, and parents have expressed concern related to the impacts of prolonged social isolation on their children’s social development [37]. These disruptions, and the associated instability, uncertainty, and social isolation, may create an added stressor related to parents’ concern for their children’s well-being and were associated with greater risk of adverse psychological outcomes [19]. Interventions geared towards safely supporting child socialization while engaging in online learning would therefore likely benefit parents and children and alleviate some of the mental health burden associated with remote learning for parents, particularly those from the most vulnerable communities.

### 4.3. Individual Risk Factors

Some sociodemographic variables including parent gender, age, and race and ethnicity appeared uncorrelated with mental health symptoms, particularly in multivariable analyses. However, higher income and education level were associated with lower odds of anxiety, depression, and PTSD, and several of these associations remained in the adjusted analysis. It is difficult to draw conclusions from the current study related to the interplay between race/ethnicity and socioeconomic status, as our sample was not evenly distributed by race/ethnicity, income, education level, and other demographic factors that may influence the observed associations. Additionally, other confounding factors including child disabilities and comorbidities have a great influence on outcomes; these limitations should be kept in mind when considering the conclusions presented. However, studies of past disasters have identified similar sociodemographic risk and protective factors on parental psychological outcomes.

Prior studies have identified marginalized race/ethnicity, low socioeconomic status, and persistent stressors including job loss as factors conferring a higher risk of negative mental health outcomes among adults following disasters, and following the COVID-19 pandemic in particular [10,14,16,38]. Studies following 9/11 and Hurricane Katrina demonstrated a higher likelihood of adverse mental health outcomes among individuals of marginalized race or ethnicity, those with lower income, those experiencing unemployment, and those experiencing social isolation following the disaster [29,39,40,41,42]. Prior work investigating the pandemic has demonstrated similar trends; studies suggest a relationship between marginalized race or ethnicity, food insecurity, housing instability, and greater stress during the pandemic [4,43]. Conversely, higher socioeconomic status has been associated with lower rates of negative parent mental health outcomes in past pandemics and during the COVID-19 pandemic [32]. Studies suggest that higher educational levels may protect against the development of anxiety and depression, with this protective impact likely mediated by socioeconomic status [42,44,45,46]. These findings highlight the complex relationships between race/ethnicity, socioeconomic status, and financial strain; these factors can contribute to and influence one another, making it difficult to isolate individual impacts. Nonetheless, our findings in the context of this prior work inform health and education policy makers by highlighting the importance of risk factors for adverse outcomes among marginalized groups and ensuring policies that support these groups and promote health and educational equity during future similar crises.

The current study also demonstrated an association between pre-existing mental health conditions and parental anxiety, depression, and PTSD symptoms during the pandemic. These associations remained in the multivariable analysis, as well as in the multivariable analysis among the subset of parents whose children engaged in remote learning. This trend was not observed for parent stress reported during the pandemic, possibly due to the overwhelming prevalence of stress in the study sample obscuring associations with individual predictors. Prior research reviewing mental health outcomes following disasters has demonstrated the major role of baseline mental health status in predicting mental illness following a disaster; established mental illness has been associated with a decreased likelihood of resilience and increased likelihood of PTSD, depression, and substance use disorders following a disaster [10]. In line with this, pre-existing mental health disorders, along with other medical problems, have been associated with negative mental health outcomes in response to COVID-19 [17,38,47]. Again, the current study highlights a vulnerable subpopulation among parents who would benefit from targeted interventions and policies to ensure adequate support, including access to counseling services and other mental health resources during the pandemic.

Notably, previous studies indicated a variation in risk factors depending on the specific adverse psychological outcome. The magnitude of disaster exposure has been suggested to be more influential in the development of PTSD, compared to depression; this is supported by the current study, in which greater disruption in children’s lives and specific routine changes were associated with higher odds of PTSD symptoms among parents [10,48]. Meanwhile, individual factors, including socioeconomic status and stressors in the aftermath of the disaster, may be more related to the development of depression [10,48]. The current study somewhat aligns with these prior findings: higher income and education level were associated with lower odds of anxiety and PTSD, in addition to depression. However, only the associations with anxiety and depression remained in multivariable analyses. Our findings may be helpful in identifying those at risk of developing certain adverse mental health outcomes and providing appropriate psychological support to improve outcomes.

### 4.4. Protective Factors: Social Support and Resilience

Among our sample, more frequent social support was associated with lower odds of parent-reported symptoms of stress, depression, and PTSD. Low social support has been associated with greater risk of negative mental health outcomes following a disaster, including symptoms of PTSD and depression; in particular, social support following Hurricane Katrina was found to protect against development of these outcomes [10,40,42,49]. During the COVID-19 pandemic, low social support has been associated with parental burnout, and lack of social resources has been associated with development of depression [14,50]. Conversely, greater support and perceived control have been associated with lower perceived parental stress and less potential for child abuse [51].

Greater resilience was also associated with lower odds of adverse parent psychological outcomes in our study. The CD-RISC2 utilized in this study assesses adaptability to change and the ability to rebound from challenges, with a higher score indicating greater resiliency; reliability and validity studies reported a baseline score of 6.91 in the general population, slightly higher than the mean score of 6 observed in our overall sample [28]. Resilience has been identified as a protective factor against development of negative mental health outcomes in reviews of past disasters, and in studies of COVID-19 [10,18,52]. Resilience, in turn, may be influenced by disaster-related factors including severity of trauma exposure, change in income, and social support, along with individual factors including gender, age, race or ethnicity, and education [52]. Individual traits and abilities including flexibility, optimism, self-efficacy, self-control, a sense of meaning, and coping skills have been associated with greater resilience and decreased likelihood of negative mental health outcomes after disasters [10,53]. The findings in the current study indicate the types of policies, programs, and interventions that may protect parents from adverse mental health outcomes during crises such as the pandemic, including those that bolster social support and build individual and community resilience.

### 4.5. Impact of Parent Mental Health on Families during the Pandemic

Prior models based on family systems’ theory describe the pandemic’s influence at the family level, with pandemic and family stressors influencing parent well-being, and, in turn, parent well-being impacting the other members of the family [54,55]. These models suggest the contributory role of pandemic stressors on caregiver burden, adversely impacting not only parent mental health but also parenting practices and parent–child relationships; studies have demonstrated associations between parent-perceived stress and the increased likelihood of child maltreatment [51,55,56]. Additionally, mental health symptoms and behavior changes among children have been associated with psychological problems among parents following disasters [57,58]. Conversely, the pandemic also has the potential to positively impact family cohesion, communication, and functioning. Parent adaptability during COVID has been associated with positive parenting practices, and parents may demonstrate effective coping skills for their children; this may provide a source of resilience for the family, and foster adaptability and positive outcomes across the family unit [59,60]. Interventions at the family level focused on providing education and resources to build family cohesion, communication, and resilience can therefore be beneficial not only in minimizing negative mental health outcomes among parents, but also among children.

### 4.6. Limitations

This is a cross-sectional analysis, and the associations we present cannot be interpreted as causal relationships. Confounding must be considered, as those who experienced the disruptions examined in this study may also be more likely to experience other factors placing them at risk of adverse mental health outcomes, including being an essential worker, lacking paid sick leave, attending crowded schools, and neighborhood segregation. Additionally, confounding participant characteristics including having children with disabilities and other comorbid conditions must be considered, as these factors can greatly influence outcomes and were not considered in this analysis. The goal of the Speak Up on COVID study was to survey a diverse adult NYC population; the survey was not designed specifically for parents and consequently had minimal inclusion or exclusion criteria. This also limited the addition of questions related to children, including those assessing child factors including developmental disabilities and other conditions. Our study sample is a convenience sample, and therefore is not generalizable to the NYC population at large as it is limited in size and is not representatively distributed across sociodemographic variables. The smaller sample size, particularly among vulnerable populations including marginalized racial/ethnic groups and those of lower educational and income level, decreases statistical power and renders multivariable results more difficult to interpret. Additionally, stress was highly prevalent among the sample, making it difficult to interpret associations between predictors and stress; stress was also measured using a one item scale, as opposed to more detailed standardized instruments, and therefore further research should be done to confirm our findings.

## 5. Conclusions

The current study highlights pandemic-related disruptions in child routines, pre-existing mental illness, and lower socioeconomic status as major factors conferring greater risk for adverse psychological outcomes among parents in NYC during the pandemic. Conversely, higher socioeconomic status, social support, and resilience emerged as factors with a protective effect against the development of adverse parent mental health outcomes. These findings align with prior studies investigating the psychological impacts of the pandemic upon parents and families, with the current study contributing an analysis of these trends among a parent population in a major epicenter of the pandemic [32,38,61]. Given the lasting influence upon child well-being and family cohesion, it is critical to assess parent mental health outcomes in the aftermath of the COVID-19 pandemic. These findings can be leveraged to advocate for the appropriate allocation of resources to provide maximal support for vulnerable populations during periods of disruption requiring remote learning and social isolation practices, as well as the allocation of mental health services to support parents through these changes. Equally important will be finding creative ways to alleviate concerns related to social isolation by allowing children to safely engage socially during the pandemic. The informed provision of resources and support, particularly among those most vulnerable to adverse psychological outcomes, will be key in mitigating the adverse impacts of the pandemic upon parents, and by extension, upon families and children. Future directions include further studies to assess the long term mental health outcomes among parents in the aftermath of the pandemic, as well as studies to better characterize mental health outcomes among children and the associations between parent and child outcomes to guide interventions and policies. Additionally, we are analyzing qualitative data to better understand the unique circumstances and individual parent experiences during the pandemic, which may provide valuable narrative information to further guide programs and resources for families.

## Figures and Tables

**Table 1 ijerph-19-06442-t001:** Primary outcomes and baseline characteristics of respondents with children under 18 Years old completing the Speak Up on COVID Survey, May 2020–April 2021, New York City.

Total Responses with Children under 18 Years Old	523
Characteristics	N (%) or Median (IQR)
Mental health symptoms	
Anxiety	214 (41.2)
Depression	137 (26.6)
PTSD	176 (33.7)
Stress	474 (91.0)
Gender	
Man	117 (22.4)
Woman	402 (76.9)
Other	4 (0.8)
Age	
18–24	82 (16.3)
25–44	256 (50.8)
45–64	155 (30.8)
65+	11 (2.2)
Race	
Non-Hispanic white	191 (36.5)
Non-Hispanic Black	77 (14.7)
Hispanic	143 (27.3)
Non-Hispanic Asian	77 (14.7)
Other/mixed	35 (6.7)
Income	
<USD 20 k	46 (9.4)
USD 20 k–USD 39,999	52 (10.7)
USD 40 k–USD 59,999	53 (10.9)
USD 60 k–USD 79,999	29 (6.0)
USD 80 k+ ^a^	246 (50.5)
Unknown	61 (12.5)
Education	
High school/GED or less	57 (11.8)
Some college	90 (18.7)
Bachelor	133 (27.6)
Postgrad	202 (41.9)
Number of children’s programs/activities cancelled ^b^	
0	65 (12.4)
1–2	124 (23.7)
3–4	210 (40.2)
5–6	97 (18.5)
7–8	27 (5.2)
Most commonly reported disruptions in child routines ^b^	
After school activities cancelled	294 (56.2)
Not able to play with other children	368 (70.4)
Not able to go outside and play	288 (55.1)
Use the internet/TV to entertain more	275 (52.6)
Remote learning	
Participated in remote learning	298 (60.3)
Feel prepared for remote learning	130 (44.7)
Overwhelmed by remote learning	206 (70.3)
Resiliency score	6 (5, 8)
Social support	
<once weekly	82 (16.1)
1–2 times weekly	122 (23.9)
3–5 times weekly	117 (22.9)
>5 times weekly	181 (35.5)
Prefer not to say	8 (1.6)
Pre-existing mental illness	156 (29.8)
Pre-existing anxiety	122 (23.3)
Pre-existing depression	86 (16.4)
Pre-existing PTSD	29 (5.5)
Pre-existing other mental illness	25 (4.8)

^a^ Indicates all reported annual incomes greater than 80,000 USD. ^b^ Disruption in child routines was assessed using the question, “How has the pandemic affected children in your home? Please check all that apply,” with a checklist including the disruptions listed in the table [26].

**Table 2 ijerph-19-06442-t002:** Associations between the impact of the pandemic upon children’s lives and parent’s mental health symptoms, unadjusted results.

	Anxiety		Depression		PTSD		Stress	
	OR ^a^ (95% CI)	*p*	OR ^a^ (95% CI)	*p*	OR ^a^ (95% CI)	*p*	OR ^a^ (95% CI)	*p*
Number of children’s programs/activities cancelled								
0	1		1		1		1	
1–2	0.80 (0.42, 1.54)	0.502	0.53 (0.26, 1.1)	0.085	1.07 (0.54, 2.17)	0.858	0.72 (0.27, 1.77)	0.495
3–4	1.30 (0.73, 2.37)	0.377	0.90 (0.49, 1.72)	0.747	1.53 (0.83, 2.95)	0.187	1.17 (0.44, 2.78)	0.740
5–6	**1.98 (1.04, 3.85) ^b^**	**0.040**	1.26 (0.64, 2.54)	0.514	**2.65 (1.35, 5.4)**	**0.006**	**11.67 (2, 221)**	**0.023**
7–8	**4.43 (1.72, 12.29)**	**0.003**	2.19 (0.85, 5.68)	0.104	**2.84 (1.11, 7.4)**	**0.030**	3.19 (0.53, 61.29)	0.289
Most commonly reported disruptions in child routines								
After school activities cancelled	1.15 (0.8, 1.63)	0.452	1.21 (0.82, 1.81)	0.341	**1.65 (1.14, 2.4)**	**0.009**	1.83 (1, 3.4)	0.050
Not able to play with other children	**1.74 (1.18, 2.61)**	**0.006**	1.19 (0.78, 1.86)	0.429	**1.54 (1.03, 2.35)**	**0.040**	1.13 (0.58, 2.12)	0.711
Not able to go outside and play	**1.80 (1.26, 2.58)**	**0.001**	**1.93 (1.29, 2.91)**	**0.002**	**1.64 (1.13, 2.39)**	**0.009**	**2.34 (1.27, 4.44)**	**0.008**
Use the internet/TV to entertain more	1.32 (0.93, 1.87)	0.124	1.27 (0.86, 1.89)	0.228	1.34 (0.93, 1.93)	0.117	**2.11 (1.14, 4)**	**0.019**
Remote learning								
Participated in remote learning	1.28 (0.89, 1.86)	0.188	0.85 (0.56, 1.28)	0.431	1.13 (0.77, 1.66)	0.526	0.71 (0.36, 1.37)	0.322
Feel prepared for remote learning	**0.44 (0.27, 0.71)**	**0.001**	**0.41 (0.23, 0.72)**	**0.003**	**0.50 (0.3, 0.82)**	**0.006**	**0.24 (0.09, 0.55)**	**0.001**
Overwhelmed by remote learning	**2.39 (1.41, 4.16)**	**0.002**	**2.03 (1.09, 4.01)**	**0.033**	**1.91 (1.11, 3.4)**	**0.023**	**3.70 (1.67, 8.36)**	**0.001**

^a^ Odds ratios were generated from univariable logistic regression models examining associations between predictors and the four outcomes. ^b^ Bold values indicate statistically significant results.

**Table 3 ijerph-19-06442-t003:** Associations between individual factors and parent mental health symptoms, unadjusted results.

	Anxiety		Depression		PTSD		Stress	
	OR ^a^ (95% CI)	*p*	OR ^a^ (95% CI)	*p*	OR ^a^ (95% CI)	*p*	OR ^a^ (95% CI)	*p*
Gender								
Man	1		1		1		1	
Woman	1.51 (0.99, 2.35)	0.062	1.42 (0.88, 2.36)	0.166	0.96 (0.63, 1.5)	0.863	1.20 (0.58, 2.32)	0.613
Other	1.97 (0.23, 16.96)	0.504	1.21 (0.06, 9.95)	0.869	1.92 (0.22, 16.53)	0.520	NA ^b^	0.98
Age								
18–24	1		1		1		1	
25–44	1.00 (0.61, 1.66)	0.994	0.66 (0.39, 1.13)	0.123	1.16 (0.69, 1.99)	0.576	1.15 (0.44, 2.74)	0.756
45–64	0.73 (0.42, 1.27)	0.269	**0.51 (0.28, 0.93) ^c^**	**0.027**	0.86 (0.49, 1.54)	0.606	0.76 (0.28, 1.84)	0.556
65+	0.28 (0.04, 1.16)	0.115	0.41 (0.06, 1.71)	0.269	0.45 (0.07, 1.91)	0.332	0.93 (0.14, 18.38)	0.951
Race								
Non-Hisp white	1		1		1		1	
Non-Hisp Black	0.84 (0.48, 1.44)	0.528	1.43 (0.77, 2.62)	0.250	0.76 (0.42, 1.35)	0.362	0.51 (0.21, 1.29)	0.142
Hispanic	1.20 (0.78, 1.87)	0.407	**1.70 (1.03, 2.81)**	**0.037**	1.56 (0.99, 2.44)	0.053	0.61 (0.27, 1.37)	0.233
Non-Hisp Asian	0.87 (0.5, 1.51)	0.632	1.66 (0.9, 3.02)	0.100	0.76 (0.42, 1.35)	0.362	**0.40 (0.17, 0.97)**	**0.039**
Other/mixed	1.39 (0.67, 2.87)	0.376	1.35 (0.56, 3.04)	0.483	0.70 (0.3, 1.54)	0.398	2.21 (0.41, 40.98)	0.453
Income								
<USD 20 k	1		1		1		1	
USD 20 k–USD 39,999	0.73 (0.33, 1.61)	0.433	1.01 (0.44, 2.32)	0.976	0.63 (0.28, 1.39)	0.252	0.90 (0.21, 3.6)	0.875
USD 40 k–USD 59,999	0.88 (0.4, 1.95)	0.757	0.98 (0.43, 2.25)	0.966	0.47 (0.21, 1.06)	0.072	1.17 (0.26, 5.21)	0.834
USD 60 k–USD 79,999	0.98 (0.39, 2.5)	0.970	0.77 (0.28, 2.04)	0.600	**0.26 (0.08, 0.73)**	**0.014**	NA ^b^	0.983
USD 80 k+	**0.50 (0.26, 0.95)**	**0.033**	**0.38 (0.19, 0.76)**	**0.006**	**0.47 (0.25, 0.9)**	**0.021**	1.13 (0.32, 3.2)	0.828
Unknown	0.62 (0.28, 1.35)	0.232	0.79 (0.35, 1.79)	0.567	**0.39 (0.17, 0.86)**	**0.021**	0.54 (0.14, 1.78)	0.332
Education								
High school/GED or less	1		1		1		1	
Some college	**0.47 (0.24, 0.93)**	**0.031**	**0.50 (0.25, 0.99)**	**0.047**	0.61 (0.31, 1.2)	0.156	0.49 (0.13, 1.49)	0.238
Bachelor	**0.44 (0.23, 0.83)**	**0.011**	**0.38 (0.2, 0.72)**	**0.003**	**0.51 (0.27, 0.97)**	**0.039**	0.55 (0.15, 1.59)	0.308
Postgrad	**0.43 (0.23, 0.78)**	**0.006**	**0.22 (0.12, 0.41)**	**0.000**	**0.46 (0.25, 0.84)**	**0.011**	1.61 (0.42, 5.16)	0.443
Resiliency score								
1 unit higher	**0.78 (0.7, 0.87)**	**<0.001**	**0.73 (0.65, 0.82)**	**<0.001**	**0.74 (0.66, 0.82)**	**<0.001**	**0.65 (0.5, 0.81)**	**<0.001**
Social support								
<once weekly	1		1		1		1	
1–2 times weekly	0.69 (0.39, 1.22)	0.204	0.78 (0.43, 1.42)	0.417	0.85 (0.48, 1.52)	0.590	0.14 (0.01, 0.74)	0.062
3–5 times weekly	0.57 (0.32, 1.02)	0.057	**0.53 (0.28, 0.99)**	**0.047**	0.71 (0.39, 1.27)	0.242	**0.12 (0.01, 0.62)**	**0.042**
>5 times weekly	0.60 (0.35, 1.01)	0.056	**0.45 (0.25, 0.8)**	**0.006**	**0.50 (0.29, 0.86)**	**0.012**	**0.09 (0, 0.44)**	**0.019**
Prefer not to say	0.71 (0.13, 3.43)	0.672	1.30 (0.24, 6.28)	0.742	0.47 (0.07, 2.19)	0.373	0.09 (0, 2.34)	0.095
Pre-existing mental illness								
No	1		1		1		1	
Yes	**3.26 (2.21, 4.84)**	**<0.001**	**3.46 (2.29, 5.23)**	**<0.001**	**4.64 (3.12, 6.96)**	**<0.001**	NA ^b^	0.984

^a^ Odds ratios were generated from univariable logistic regression models examining associations between predictors and the four outcomes. ^b^ The number of observations in some categories was too low, resulting in an inflated OR. ^c^ Bold values indicate statistically significant results.

## Data Availability

The data presented in this study are available in Table 1, Table 2 and Table 3, Appendix A.

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
