# Peer review of "Child Disruptions, Remote Learning, and Parent Mental Health during the COVID-19 Pandemic"

_ijerph, 2022, doi:10.3390/ijerph19116442_

Round 1

Reviewer 1 Report

The manuscript under review attempts to evaluate the Impact of Child Disruptions and Remote Learning on Parent 2 Mental Health during the COVID-19 Pandemic. In general, the manuscript captures details of the study design and implementation of the project. All the sections of the manuscript are well written and concluded, although the limitations are fewer, and it has been presented in the manuscript. The study is of sound design and has a clear practical and clinical interest. find below the minor comments 

Abstract:

  • NYC  - provide the complete form once,  when it appears first
  • Did the authors use MesH keywords?

Introduction:

  • 4th paragraph Line 68-77 is repetition
  • Kindly write briefly about remote learning, online education
  • Refer to the below-mentioned article and cite it accordingly

Soltaninejad M, Babaei-Pouya A, Poursadeqiyan M, Feiz Arefi M. Ergonomics factors influencing school education during the COVID-19 pandemic: A literature review. Work. 2021 Jan 1;68(1):69-75.

Starkey L, Shonfeld M, Prestridge S, Cervera MG. Covid-19 and the role of technology and pedagogy on school education during a pandemic. Technology, Pedagogy and Education. 2021 Jan 1;30(1):1-5.

Materials and Methods

  • We piloted the survey with partners and translated it into ten languages commonly spoken in NYC. The survey was finalized and launched in May of 2020:- is the survey validated? If yes, Provide details (number, approving authority). Published?
  • 3. Predictors: suggested providing a table instead explaining in text. Which will be more presentive and easy for the readers
  • How did the authors obtain the participant's consent? Was it included in the online questionnaire? Mention details
  • Kindly mention the ethical approval details at the start of materials and methods and later in the declaration section, which the authors already provide

Results:

  • Results are well written and presented; few minor comments
  • Kindly shift the tables to the places were cited in the main text; for example, Table 1 to line 179 or 186
  • Avoid repetition of results from tables in the text. Tables showing all the results advised not to write in text.

Discussion:

  • Discuss the possible interventions and how the research outcome will be helpful for the children and parents in each section of the discussion
  • How do the current study results help the education and health sector's policies during the Pandemic?
  • Limitations were presented; kindly provide the future directions or recommendations

Author Response

Reviewer 1 Comments:

Abstract:

  • NYC - provide the complete form once, when it appears first

Line 22: Provided complete form.

  • Did the authors use MesH keywords?

Lines 40-41: Revised keywords to include MesH keywords.

Introduction:

  • 4th paragraph Line 68-77 is repetition

Lines 79-94: This paragraph was intended to describe the mental health burdens of the pandemic upon parents, specifically. We revised this paragraph to make it less redundant by adding the key points to the previous paragraph.

  • Kindly write briefly about remote learning, online education
  • Refer to the below-mentioned article and cite it accordingly
    • Soltaninejad M, Babaei-Pouya A, Poursadeqiyan M, Feiz Arefi M. Ergonomics factors influencing school education during the COVID-19 pandemic: A literature review. Work. 2021 Jan 1;68(1):69-75.
    • Starkey L, Shonfeld M, Prestridge S, Cervera MG. Covid-19 and the role of technology and pedagogy on school education during a pandemic. Technology, Pedagogy and Education. 2021 Jan 1;30(1):1-5.

Lines 61-66: Added additional background information about remote learning to the existing background info that focused on remote learning in NYC specifically. We cited both suggested articles.

Materials and Methods:

  • We piloted the survey with partners and translated it into ten languages commonly spoken in NYC. The survey was finalized and launched in May of 2020:- is the survey validated? If yes, Provide details (number, approving authority). Published?

Lines 114-116: The goal of this survey was to examine a broad range of outcomes of COVID in communities within NYC that were disproportionately affected by the pandemic. The survey was composed of a variety of domains that were of interest to researchers and community partners assessing these different outcomes. While we did not validate the survey in its entirety given the context of the pandemic and the rapid timeline to develop and deploy the survey, we did choose validated questions and scales for most outcomes. A detailed breakdown of the sources of the survey questions are available upon request or can be provided as a supplemental table if needed. We included references in the Predictors section of the manuscript for several of the surveys that we utilized, including a parenting survey from the University of Michigan for questions assessing disruption in routines (Reference 26, cited on line 154), the UCSF Eureka survey assessing prior psychological history (Reference 27, cited on line 175), and the CD-RISC2 scale used to assess resilience (Reference 28, cited on line 178).

  • Predictors: suggested providing a table instead explaining in text. Which will be more presentive and easy for the readers

Lines 153, 170, and 181: Provided in-text citations to data table 1, which presents predictors in a table.

  • How did the authors obtain the participant's consent? Was it included in the online questionnaire? Mention details

Line 131: Added statement explaining that the study was IRB exempt, full consent was not required and instead we presented an IRB approved research information sheet with study details, which all participant had to review prior to participation and attest that they met eligibility criteria. Form attached and sent to Mr. Liu.

  • Kindly mention the ethical approval details at the start of materials and methods and later in the declaration section, which the authors already provide

Line 107: Added statement of ethical approval details at the start of the section.

Results:

  • Results are well written and presented; few minor comments
  • Kindly shift the tables to the places were cited in the main text; for example, Table 1 to line 179 or 186

Lines 205, 239, and 284: Shifted tables accordingly.

  • Avoid repetition of results from tables in the text. Tables showing all the results advised not to write in text.

Lines 210-231: Edited results to avoid repetition of results included in tables.

Discussion:

  • Discuss the possible interventions and how the research outcome will be helpful for the children and parents in each section of the discussion
  • How do the current study results help the education and health sector's policies during the Pandemic?

Lines 337-341, 362-366, 374-377, 410-413, 426-429, 440-442, 465-468, 483-486: Discussed possible interventions and contributions of his study for children and parents, and how the current findings can inform policies during the pandemic.

  • Limitations were presented; kindly provide the future directions or recommendations

Lines 528-534: Discussed future directions and recommendations based on the current findings.

Reviewer 2 Report

This is an interesting study about how child disruption impact parent mental health during COVID-19 pandemic. The authors collected data from NYC across more than 100 community, the relationship between parents’ anxiety, depression, post-traumatic stress disorder, stress and their children’s activity were analyzed.

There is one big concern for the statistical analysis in this manuscript. From the table listed here, reader cannot find how did author get the odds ratio, cannot make sure whether the strategy used in analysis is right or wrong. Data listed in the table missing important message for odds ratio calculation.

Author also stated univariable and multivariable logistic regression models, what’s the strategy?

Author Response

Reviewer 2 Comments:

  • There is one big concern for the statistical analysis in this manuscript. From the table listed here, reader cannot find how did author get the odds ratio, cannot make sure whether the strategy used in analysis is right or wrong. Data listed in the table missing important message for odds ratio calculation.
  • Author also stated univariable and multivariable logistic regression models, what’s the strategy?

Lines 241 and 285: Included footnote with tables outlining that the OR’s were obtained through logistic regression models assessing associations between all predictors and the four outcomes.

Reviewer 3 Report

Dear Authors, 

I appreciated reviewing this important and interesting paper on the impact of pandemic stressors on the mental health of parents. I believe this is a very interesting issue that needs to be empirically investigated since the examination of potential stressors may help both researchers and practitioners to contain the effects of the pandemic, by providing knowledge about specific areas that should be addressed promptly. In addition to this, this study may also shed new light on specific populations that may be more vulnerable and exposed to major risks of undermining mental health.

Despite the relevance of this work, there are some main concerns that should be considered and addressed before its official publication in the International Journal of Environmental Research and Public Health.

Abstract

  • I would suggest the authors provide a few words of context before the description of what has been conducted in this work
  • I would also suggest the authors use the impersonal style in the abstract, instead of the “we” form

Introduction

  • At the end of the introduction, the main goals and specific hypotheses of this work should be added. Are there any specific hypotheses the authors are expected to consider regarding these stressors on mental health? This part should be specified and deepened more.

Materials and Methods
Speak up on COVID Survey

  • Why did the authors consider convenience sampling instead of conducting a power analysis before starting collecting data? This might have increased the power of the study
  • Are there any inclusion or exclusion criteria in the selection of the sample? Having a child under 18 years of age at home is vague. For example, are considered children with developmental disabilities, neurodevelopmental disorders, and others? I truly believe that if also neuroatypical conditions are included in the sample this factor should be considered.
  • I would add here a paragraph “participants” with all the information about the participants and recruitment, otherwise, information is scattered throughout the text and difficult to follow. Also, the information about the sample size is only provided in the results session and should be reported before

Outcomes

  • The stress variable may be not reliable since it’s just based on one item/answer. I suggest the authors take into consideration the idea of investigating stress with standardized instruments (e.g., Parental Stress Index - PSI) that can provide a more reliable measure of parental stress. I think the authors should state at some point that further research needs to be done especially considering this variable that needs to be confirmed by other standardized instruments

Predictors

  • How is child disruption measured? Are there specific questions measuring for example cancellation of after-school activities? How are these questions formulated? I would suggest the authors provide examples and more explanations about this
  • Line 138: I would suggest the authors explain better this aspect. What does it mean “indicate that a doctor or nurse had told them they had anxiety, depression, PTDS, or another mental health condition” - On which basis the doctor/ nurse told it? More explanation should be provided here

Statistical Analysis

  • Are the variables checked for normality and linearity before the implementation of univariate and multivariable logistic regression models?
  • Line 150: Still not clear to me how the authors measured disruptions in child routines and which type of variable is this.
  • How is multicollinearity checked? Please specify this

Results

  • As already mentioned, I would suggest authors create a paragraph before with the information considering the participants

Discussion

  • In general, the whole discussion should be moderated. There are some variables and factors (e.g., presence of disabilities or disorders) that may explain the data or part of them so conclusions should be drawn with caution

Author Response

Reviewer 3 Comments:

Abstract:

  • I would suggest the authors provide a few words of context before the description of what has been conducted in this work

Lines 22-23: Added context before discussing this work.

  • I would also suggest the authors use the impersonal style in the abstract, instead of the “we” form

Lines 22-26: Changed “we” to an impersonal style.

Introduction:

  • At the end of the introduction, the main goals and specific hypotheses of this work should be added. Are there any specific hypotheses the authors are expected to consider regarding these stressors on mental health? This part should be specified and deepened more.

Lines 100-104: Added more details about the specific goals and hypotheses of this work.

Materials and Methods:

Speak up on COVID Survey:

  • Why did the authors consider convenience sampling instead of conducting a power analysis before starting collecting data? This might have increased the power of the study

Lines 127-130: Included statement explaining our decision to use convenience sampling: the goal of the Speak Up on COVID survey was to understand the broad range of impacts of the pandemic on NYC residents who were disproportionately affected by COVID. In order to achieve this goal, we chose a sampling strategy based on outreach through our community partners and targeted outreach in neighborhoods most impacted by COVID. This resulted in our convenience sample and the descriptive analyses we conducted. As there was no intervention and no specific outcome of interest, we had basis for a power analysis. We mention that, as a convenience sample, our study is not generalizable to the NYC population at large in line 521.

  • Are there any inclusion or exclusion criteria in the selection of the sample? Having a child under 18 years of age at home is vague. For example, are considered children with developmental disabilities, neurodevelopmental disorders, and others? I truly believe that if also neuroatypical conditions are included in the sample this factor should be considered.

Lines 512-518: Included a statement about the study population and reason for minimal exclusion criteria. This study had minimal inclusion or exclusion criteria because we wanted to collect data from a diverse sample adult NYC residents. When developing the Speak Up on COVID survey we had to balance the many outcomes of interest with survey length and participant burden. Some members of our team were specifically interested in impacts on children and families, so we included questions to examine these outcomes. However, the survey was not designed for parents specifically and this limited our ability to ask detailed questions about children's developmental disabilities or other conditions. We agree that this data is important and would have been interesting and relevant, but in creating the survey we had to prioritize survey items based on the feedback of the many academic and community partners involved. This is a study limitation that we discussed in lines 512-518 as mentioned above.

  • I would add here a paragraph “participants” with all the information about the participants and recruitment, otherwise, information is scattered throughout the text and difficult to follow. Also, the information about the sample size is only provided in the results session and should be reported before

Lines 119-136: Added a paragraph with all information about participants and recruitment to organize this information. Also reported the information regarding sample size.

Outcomes:

  • The stress variable may be not reliable since it’s just based on one item/answer. I suggest the authors take into consideration the idea of investigating stress with standardized instruments (e.g., Parental Stress Index - PSI) that can provide a more reliable measure of parental stress. I think the authors should state at some point that further research needs to be done especially considering this variable that needs to be confirmed by other standardized instruments

Lines 527-529: Added a statement about the limited reliability of this measure of stress, and stating the need for further research particularly characterizing stress with more reliable measures. 

Predictors:

  • How is child disruption measured? Are there specific questions measuring for example cancellation of after-school activities? How are these questions formulated? I would suggest the authors provide examples and more explanations about this

Line 153: Added a statement explaining this item and included the question used to assess it, with the reference to the survey that we adapted this question from. We also included the survey question as a footnote in table 1 (line 207).

  • Line 138: I would suggest the authors explain better this aspect. What does it mean “indicate that a doctor or nurse had told them they had anxiety, depression, PTDS, or another mental health condition” - On which basis the doctor/ nurse told it? More explanation should be provided here

Line 172: Added a statement explaining this question, and referenced the validated source it was adapted from.

Statistical Analysis:

  • Are the variables checked for normality and linearity before the implementation of univariate and multivariable logistic regression models?
  • How is multicollinearity checked? Please specify this

Line 184: Included a statement confirming this.

  • Line 150: Still not clear to me how the authors measured disruptions in child routines and which type of variable is this.

Lines 153 and 207: As mentioned above, we included the survey question used to assess this item as well as the original survey it was adapted form in a footnote for table 1.  

Results:

  • As already mentioned, I would suggest authors create a paragraph before with the information considering the participants

Lines 119-136: As mentioned above, added a paragraph with all information about participants and recruitment to organize this information.

Discussion:

  • In general, the whole discussion should be moderated. There are some variables and factors (e.g., presence of disabilities or disorders) that may explain the data or part of them so conclusions should be drawn with caution

Lines 396-401, 417-419, 507-518: Included statements to moderate the discussion by describing confounding factors that may account for some of the findings and therefore must be considered; we highlight that the associations in the study do not represent causal relationships and that conclusions should be drawn with caution considering these limitations.

Round 2

Reviewer 2 Report

Please adjust the table a bit, some of the numbers of 95% CI are not in a line.

No further comments.

Reviewer 3 Report

Dear authors, 

I appreciated reviewing this important and interesting paper on the impact of pandemic stressors on the mental health of parents. My comments and suggestions have been adequately answered and addressed, so I recommend this work for publication in the International Journal of Environmental Research and Public Health.